# Potential Beneficial Actions of Fucoidan in Brain and Liver Injury, Disease, and Intoxication—Potential Implication of Sirtuins

**DOI:** 10.3390/md18050242

**Published:** 2020-05-05

**Authors:** Jasmina Dimitrova-Shumkovska, Ljupcho Krstanoski, Leo Veenman

**Affiliations:** 1Department of Experimental Biochemistry, Institute of Biology, Faculty of Natural Sciences and Mathematics, University Ss Cyril and Methodius, Arhimedova 6, P.O. Box 162, 1000 Skopje, Macedonia; lkrstanoski@gmail.com; 2Israel Institute of Technology, Faculty of Medicine, Rappaport Institute of Medical Research, 1 Efron Street, P.O. Box 9697, Haifa 31096, Israel

**Keywords:** fucoidan, P-selectin, sirtuin 3, brain injury and disease, excitotoxicity, inflammation, neurodegeneration, liver injury, atherosclerosis

## Abstract

Increased interest in natural antioxidants has brought to light the fucoidans (sulfated polysaccharides present in brown marine algae) as highly valued nutrients as well as effective and safe therapeutics against several diseases. Based on their satisfactory in vitro antioxidant potency, researchers have identified this molecule as an efficient remedy for neuropathological as well as metabolic disorders. Some of this therapeutic activity is accomplished by upregulation of cytoprotective molecular pathways capable of restoring the enzymatic antioxidant activity and normal mitochondrial functions. Sirtuin-3 has been discovered as a key player for achieving the neuroprotective role of fucoidan by managing these pathways, whose ultimate goal is retrieving the entirety of the antioxidant response and preventing apoptosis of neurons, thereby averting neurodegeneration and brain injuries. Another pathway whereby fucoidan exerts neuroprotective capabilities is by interactions with P-selectin on endothelial cells, thereby preventing macrophages from entering the brain proper. Furthermore, beneficial influences of fucoidan have been established in hepatocytes after xenobiotic induced liver injury by decreasing transaminase leakage and autophagy as well as obtaining optimal levels of intracellular fiber, which ultimately prevents fibrosis. The hepatoprotective role of this marine polysaccharide also includes a sirtuin, namely sirtuin-1 overexpression, which alleviates obesity and insulin resistance through suppression of hyperglycemia, reducing inflammation and stimulation of enzymatic antioxidant response. While fucoidan is very effective in animal models for brain injury and neuronal degeneration, in general, it is accepted that fucoidan shows somewhat limited potency in liver. Thus far, it has been used in large doses for treatment of acute liver injuries. Thus, it appears that further optimization of fucoidan derivatives may establish enhanced versatility for treatments of various disorders, in addition to brain injury and disease.

## 1. Introduction

The increasing prevalence of multiple chronic diseases has triggered massive exploration of novel pharmaceuticals for their proper management [1,2]. However, therapies are often accompanied by side effects. Given that the impaired homeostasis of reactive oxygen species (ROS) is often the source for these illnesses, extensive efforts are being made to produce synthetic antioxidants that are beneficial and lack undesired side effects [3,4]. In addition, research for natural antioxidants has led to the promotion of macro algae (more commonly known as seaweeds) as natural remedies against various disorders, due to their rich content of biologically active compounds [5,6]. Initial research concerning the in vitro antioxidant properties of bioactive compounds from different classes of algae together with analyses conducted using cell cultures revealed that a fucose containing a sulfated polysaccharide, fucoidan, showed satisfactory results concerning radical scavenging, chelating properties, and lipid peroxidation inhibiting potential, which further endorsed them as potential therapies for diseases associated with ROS overproduction [7,8]. Moreover, recent studies mentioned the relevance of the sulfate content as well as molecular weight of the isolated fucoidans for the its beneficial antioxidant features. Thus, research is ongoing to optimize isolation and extraction [9,10]. Furthermore, the dissimilarities of the chemical structure of high yielding fucoidan sources, such as marine brown algae *Fucus vesiculosus*, *Undaria pinnatifida*, or *Laminaria japonica*, additionally promoted investigations regarding the effects of fucoidans with different structural characteristics [10,11]. Various research endeavors also have led to the commercial production of fucoidan food supplements. In this review, we also briefly note our preliminary results regarding antioxidant effects of fucoidan and some of its derivatives.

The encouraging results about the biological activity of fucoidan have incited screening of in vivo antioxidant and therapeutic properties of this algal polysaccharide. At the outset, utilizing the anti-inflammatory, anti-proliferative, pro-apoptotic, cytotoxic, antifungal, antiviral, and antibacterial features of fucoidan, it was used as a therapy for malignancies and organ damage in animal models [12]. The above characteristics and effects are described in further detail in this review in correlation with pre-clinical and translation research for treatment of mental disorders, and brain damage due to disease and injury. Indeed, there is substantial evidence regarding its protective and beneficial actions in the central nervous system (CNS), both at whole organ and cellular levels, which is a plausible starting point for developing novel therapies for severe neurodegenerative and neurocognitive disorders such as Alzheimer Disease (AD), Parkinson Disease (PD), and others [13,14]. Furthermore, another line of research confirmed the beneficial influence of fucoidan in preserving cellular integrity and inhibition of fibrosis in drug-induced liver injury and hepatocellular carcinoma [15,16]. In addition to brain and liver disorders, fucoidan has been used as an effective therapy against ulcerative colitis, Crohn’s disease, and arthritis [17,18]. These curative effects, together with the non-toxic, biocompatible nature of fucoidan, provides a solid basis for its preclinical and translational research.

## 2. In Vitro Antioxidant Activity of Fucoidan from Marine Algae and Commercial Supplements

Very encouragingly, fucoidan has shown satisfactory results concerning in vitro scavenging and reducing and antioxidant potentials [19]. Since the areal of distribution of the brown algae *F. vesiculosus*, *U. pinnatifida*, or *L. japonica* is restricted to various separated regions distributed worldwide, efforts are being made to uncover additional natural sources (algae species) suitable for exploitation, including extraction of satisfactory amounts of biologically active polysaccharides and polyphenols from the species in question [20]. A search is on for “optimal fucoidan sources,” with the idea to expand the present pool of commercially available supplements, and add potent prophylactics [21]. Differences in the antioxidant potency, and sulfate or polyphenol content, of the various fucoidan extracts can depend on the type and/or quality of algal species used, the different extraction methods, methods of fractionation of extracts, and purity and yield of the polysaccharides attained. Sulfate content vs. molecular weight and uronic acid percentage of the isolated polysaccharides also contribute to alteration in their antioxidant profiles [22,23]. These reported observations emphasize the importance for the analysis of the in vitro antioxidant profiles of commercially available seaweeds and comparing them with fucoidan supplements.

In general, it is accepted that the seaweed fucoidan is a modest 2.2 diphenyl-1-picrylhydrazyl (DPPH) inhibitor, when compared to radical scavenging rates of ascorbic acid or other synthetic antioxidants [23]. Studies have indicated that a high sulfate content does not necessarily invoke high DPPH quenching and established the relevance of the monosaccharide distribution as a contributing factor [24,25]. High inhibitory concentrations (IC_50_) of fucoidan supplements manufactured by Marinova (SupaFuco), together with crude fucoidan extracted from purple laver (*Porphyra sp.)* and nori (*P. tennera*) (ranging from 2.5 ± 0.18 mg/mL for SupaFuco and 12.59 ± 1.13 and 22.54 ± 2.68 mg/mL for purple laver and nori; summarized in Table 1), also endorsed these claims (own unpublished data). High polyphenol and chelating properties of purple laver significantly contribute to the DPPH neutralization rates (own unpublished data). Fortunately, according to the Food and Drug Administration (FDA), fucoidan has been recognized as safe ingredient with no side effects; the treatment should be able to intensify its primary antioxidant response in vivo by application of large doses of it [26]. 

There is consensus that fucoidan also possesses nitric oxide (NO) scavenging capabilities, but little information is available. For example, it was reported that fucoidan from Sigma Aldrich exhibits high NO scavenging potential [21]. In another study, fucoidan isolated from *S. polycystum,* showed complete neutralization of NO free radical at concentration of 1 mg/mL [27]. A recent study by us, comparing the in vitro antioxidant profiles of isolated fucoidan from commercially obtained algae, versus fucoidan supplements, acknowledged that commercial dietary supplements were relatively potent in inhibiting difenylpicrylhydrazine (DPPH) radicals and NO radical scavengers (own unpublished data). In particular, pills containing fucoidan manufactured by Marinova Ltd. showed approximately two times and 10 times lower IC_50_ values than the algae of purple laver and wakame, respectively (own unpublished data, summarized in Table 1). Furthermore, while lower than the supplements from Marinova, Daiso fucoidan supplements still present higher antioxidant activity than extracted polysaharides, independent of low sulfate content. 

In parallel, isolated fucoidans from the algae shown significantly higher radical scavenging activity for superoxide radicals (IC_50_ values of 2.29 ± 0.61 for wakame and 2.80 ± 0.33 mg/mL, reached for nori algae, summarized in Table 1), which demonstrated that the isolated fucoidan from commercially procured algae are relative efficient scavengers of moderately potent oxidants (own unpublished data)**.** Nonetheless, it was found that fucoidan supplement from Marinova Ltd. showed the highest scavenging rates for this super oxide radical (IC_50_ value 1.41 ± 0.38), with approximately two times lower IC_50_ value than both of the abovementioned algal species (wakame and nori). Interestingly, investigations of Qu et al. confirmed higher superoxide scavenging activity of crude fucoidans obtained from *L. japonica* and *E. maxima* than those from ascorbic acid used as standard [28]. The effectiveness of Marinova Fucoidan may simply relate to its high fucoidan content (25%) (Table 1). Thus, it appears that anti-oxidant activities of fucoidan and their derivatives can be further optimized. 

The established positive correlation between the superoxide (O_2_^−^) scavenging activity and the sulfate content of the polysaccharides of all the investigated specimens (Table 1, own unpublished data), corroborated other studies [22,29]. High fucoidan contents of commercial supplements followed by wakame, could be crucial for their high (O_2_^−^) scavenging rates (own unpublished data). Adequate hydroxyl radical (OH^−^) scavenging abilities were also published regarding marine algae polysaccharides, appointing that multiple factors such as the reducing power, sulfate or the polysaccharide content could contribute to the OH^−^ free radical inhibition [10,22]. Taken together, the in vitro antioxidant potentials encouraged pharmacologic studies about fucoidan’s prophylactic properties.

## 3. Digestion and Absorption of Fucoidan

In the light of the positive, curative effects of fucoidans on various form of brain disorders it is becoming interesting to gain further insights in the behavior of fucoidans in the organism. Relevant for clinical application, few studies have been done to determine the efficiency of uptake of fucoidan via the digestive system. It can be assumed that the complexity of the fucoidan structure also affects its permeability and absorption rates through the gastrointestinal tract, mostly due to the limited enzymatic potentials of the human organism. In accordance with this hypothesis, earlier studies considered the sulfated polysaccharide molecule with high molecular weight and rich content of dietary fiber as “indigestible” given the partial in vitro degradation of the chemical constituents of different brown algae such as laminarans to monosaccharide units, and complete resistance to digestion of sulfated fucans and alginates by human fecal bacteria [30]. On the other hand, ELISA quantification of fucoidan levels in blood circulation, as a more reliable method for determining the digestibility of this molecule, showed that after a 12-day dosing of galactofucan capsules from *U. pinnatifida* to human individuals, only 0.6% persisted in plasma [31]. High urinary levels of fucoidan and low permeability coefficient concerning its transport across Caco-2 cells reported by newer studies also support the findings about low uptake of fucoidan by the gastrointestinal system [32,33]. With regards to the mechanism of absorption of fucoidan, a study in 2014 suggested the involvement of the round mononuclear cells of the jejunum given the higher concentration of fucoidan observed by immunochistochemistry methods in rats fed standard chow containing 2% fucoidan for one or two weeks. Increased accumulation of fucoidan in the sinusoidal non-parenchymal cells together with the Kupffer cells also indicated these as crucial players in the internalization of fucoidan in the liver [32]. *N*-butyl-*N*-(4-hydroxybutyl) nitrosamine (BBN) was found to enhance fucoidan uptake in the small intestine and liver [32]. Since its intracellular transport hardly can be completed by simple diffusion, it is highly possible that transporters are involved in its cellular influx. Researchers predict that the sodium glucose transporter (SGLT2) and/or the glucose transporter (GLUT2) might be involved in fucoidan uptake over the blood-brain barrier (BBB), given their affinity to polysaccharides and phenols with sugar substitutes [32,34,35]. The latest research has confirmed that fucoidan enters in Caco-2 cells through clathrin-mediated endocytosis because its influx is modified by chemical inhibitors of this process [36]. 

## 4. Molecular Biological Pathways Modulated by Fucoidan

To stipulate very briefly, fucoidan appears to interact with selectin on endothelial cells, preventing leukocytes from entering the brain from the blood vessels via the blood brain barrier (BBB), i.e., closing the BBB. This of course reduces inflammatory responses inside the brain tissue proper. Intracellularly, fucoidan interacts with sirtuin 3 (SIRT3) in brain cells, which in turn modulates mitochondria activity and cell nuclear gene expression, which, for example, reduces oxidative stress, ROS generation, and mitochondrial apoptosis (via mitochondrial activity modulation), and inflammatory responses, regeneration, angiogenesis, and wound healing (via cell nuclear gene expression modulation). This will be discussed in the following sections in more detail below.

### 4.1. Fucoidan―Selectin Interactions

Fucoidan binds to P-selectin with high affinity and exerts antagonism of selective function [37]. P-selectin is found on the cell surface of endothelial cells of the BBB. It binds to glycoprotein on the cell surface of leukocytes; P-selectin is involved in rolling and arresting leukocytes on the endothelium prior to leukocyte migration into the extravascular space [38,39,40,41]. Thus, the permeability of the BBB for leukocytes can be affected by P-selectin; for example, enhanced levels of P-selectin leads to a higher BBB permeability, while decreased levels of P-selectin leads to lower permeability of the BBB. Thus, briefly, by binding to P-selectin on endothelial cells of the BBB, fucoidan can inhibit the entry of leukocytes into the brain proper, and thereby, reduce the inflammatory response [37].

### 4.2. Fucoidan―Sirtuin 3 Interactions

Another major pathway for fucoidan to ameliorate brain damage due to injury and disease appears to be its interaction with SIRT3 (Figure 1). SIRT3 is a protein that in humans is encoded by the *SIRT3* gene (sirtuin (silent mating type information regulation 2 homolog) 3 (*S. cerevisiae*)) [42]. SIRT3 is a member of the mammalian sirtuin family of proteins. SIRT3 exhibits NAD+ dependent deacetylase activity. The human sirtuins present an interesting range of molecular functions and have emerged as important proteins in aging, stress resistance, and metabolic regulation. In addition to protein deacetylation, studies have shown that the human sirtuins may also function as intracellular regulatory proteins including mono ADP ribosyltransferase activity.

Endogenous SIRT3 is a soluble protein located in the mitochondrial matrix (Figure 1) [43]. Overexpression of *SIRT3* in cultured cells increases respiration and decreases the production of ROS. Interestingly, there is a strong association between SIRT3 alleles and longevity in males. In addition to controlling metabolism at the transcriptional level, sirtuins also directly control the activity of metabolic enzymes. The presence of the sirtuin deacetylase SIRT3 in the mitochondrial matrix suggests the existence of lysine acetylated mitochondrial proteins. Indeed, SIRT3 deacetylates and activates the mammalian mitochondrial acetyl-coA synthetase (AceCS2). Furthermore, SIRT3 and AceCS2 are found complexed with one another, suggesting a critical role for control of AceCS2 activity by SIRT3 [43].

In addition to its reported mitochondrial function, some researchers have proposed a very small pool of active SIRT3 exists in the cell nucleus (Figure 1). This pool is reported to consist of the long form of SIRT3 and has been suggested to have histone deacetylase activity [44]. The observation that SIRT3 has cell nuclear activity came from a report that SIRT3 protected cardiomyocytes from stress mediated cell death and that this effect was due to deacetylation of a nuclear factor, Ku-70 [45]. Presently, not many brain research studies have linked Fucoidan with SIRT3, thus, the question of potential fucoidan-SIRT interactions is worthwhile to address. The most clear-cut fucoidan-SIRT interaction is established in a traumatic brain injury (TBI) model, see Section 5, here below.

## 5. Fucoidan and Traumatic Brain Injury (TBI)

As said, with regard to brain damage, to date there is only one study of brain injury and neurodegeneration directly targeting the link between fucoidan and SIRT3.

Depending on its severity, TBI can lead to death and/or disabilities. Outcomes of TBI are mainly characterized by disturbances in the normal physiology and structure of the brain caused by extrinsic mechanical insults due to assault or other accidents [46,47]. Mechanical forces can inflict direct neuronal and astrocytic death, axonal degeneration, and vascular damage, also known as primary injuries [48]. Subsequently, the death and damage of neurons and activation of astrocytes are accompanied by additional systemic and intracranial complications, typically involving microglial activation, cytokine release, oxidative stress, and inflammation, which eventually lead towards apoptosis and necrosis [49]. This heterogenic and diverse nature of human TBI, together with the limited success of bringing adequate treatment to the patients, further emphasizes the need for broadening the research for efficient yet safe therapeutic treatments for this condition [50].

Earlier reports about successful implementation of fucoidan treatment for cardiac dysfunction or renal ischemia reperfusion injury, disorders with similar molecular response as TBI, encouraged the exploitation of this polysaccharide component as efficient therapy [51,52]. Regarding TBI itself, Wang et al. conducted a thorough study concerning the effects of this novel therapeutic as prevention or treatment for brain injuries and discovered that low molecular weight fucoidan (LWMF) at doses of 10 and 50 mg/kg significantly reduced both cortical and hippocampal lesion volume [53]. Importantly for clinical considerations, LMWF was effective even when administered up to 4 h after TBI. Given prior to TBI, fucoidan prevented contusion injuries and tissue loss in the cortex and hippocampus, which was associated with positive outcomes of behavioral tests [53]. This protection was associated with reduced neuronal apoptosis, as evidenced by TUNEL staining. Moreover, administration of fucoidan significantly reduced oxidative stress as confirmed by the decreased levels of MDA, 4-hydroxynonenal (4-HNE), protein carbonyls levels, and ROS levels, as well as reversed glutathione peroxidase (GPx), catalase (Cat), and SOD activity accompanied by restoration of mitochondrial cytochrome c levels [53] (for illustration, see Figure 1).

Finally, and importantly to understand better the workings of fucoidan, the authors reported significantly elevated levels of SIRT3 after TBI (Figure 1). The expression of SIRT3 was detected by RT-PCR and Western blot. SIRT3, as mentioned above, is a protein involved in the activation of enzymatic ROS quenching mechanisms. Fucoidan treatment of TBI further enhanced SIRT3 levels, which might be part of one of the regulatory pathways that triggered the restoration of GPx, SOD, and Cat levels [53]. Application of intracerebroventricular injection of small interfering RNA (siRNA) to induce knockdown of SIRT3 partially prevented the therapeutic effects of LMWF. Another study presented similar results after administration of commercial fucoxanthine in in vitro and in vivo TBI models, further revealing that these protective effects are only enabled through Nrf 2-induced activation of the antioxidant response element (ARE) [54]. In summary, it appears that fucoidan achieves its neuroprotective role via “triple impact.” Namely, rather than depending solely on its own chelating and radical scavenging properties, fucoidan further protects neuronal integrity by stimulating important genetic/molecular pathways capable of retrieving the entirety of the cellular antioxidant mechanisms. Furthermore, as mentioned, by interacting with P-selectin, fucoidan prevents the entry of leukocytes from the blood stream into the brain tissue proper (Figure 2).

## 6. Fucoidan and Neurodegeneration

Neurodegeneration is commonly defined as progressive atrophy and loss of function of neurons reflected in neurodegenerative diseases such as AD or PD [55]. In particular, neurodegenerative disorders are characterized by a progressive decline of motor and/or cognitive functions caused by the selective degeneration and loss of neurons within the CNS [56]. While substantial progress has been made in revealing the cellular and molecular pathways causing these conditions, which mostly involve dysregulation of genetic expression indubitably accompanied by synthesis of truncated protein triggering neuroinflammatory response [57], they still pose a significant threat for human health in general, mainly because of their severe and aggressive symptomatology, and a pathophysiology that is still a diagnostic challenge [58]. For treatment, or rather, for the alleviation of symptoms, AD patients mostly rely on cholinesterase inhibitors, while PD is mainly treated with dopamine precursors; however, to this date, no decisive progress achieved AD, PD, and other neurodegenerative disease, including neurodegeneration following brain injury [59]. Giving hope, however, favorable results in treatment of neurodegeneration have been reported with fucoidans, fucoxanthins, and other bioactive compounds from various algal sources, associated with characteristics such as: (i) scavenging potentials, which can avert neuronal damage given the linkage of this condition to neuronal ROS imbalance caused by mitochondrial dysfunction [60]; (ii) reported acetylcholinesterase (AcHE) or butyrilcholinesterase (BcHE) inhibitory activity of polyphenol rich extracts [61,62]; (iii) reported beta secretase (BACE-1) inhibitory activities of crude extracts and polysaccharides, which may preclude amyloid beta (Aβ) accumulation [61,63]. (see also Figure 2)

Newer reports about the molecular pathways involved in fucoidan neuroprotective effects on dopaminergic nerve precursor cells (MN9D) treated with 1-methyl-4-phenyl pyridine (MPP^+^) suggest its involvement in increasing superoxide dismutase (SOD) activity and reduced glutathione (GSH) concentration and decreasing the apoptosis levels by downregulation of Bax expression [13]. A study by Zhang et al. demonstrated that acute high dose co-treatment with fucoidan isolated from *L. japonica* significantly reduced rotenone-induced loss of substantia nigra pars compacta and striatal neurons [14]. This resulted in a significant improvement of the animals’ behavior and alleviation of PD symptoms by increasing mitochondrial respiratory function together with the inhibition of malondialdehyde (MDA), 8-hydroxy-2-deoxyguanosine (8-OHdG), and 3-nitrotyrosine (3-NT) formation in rat ventral midbrain [14]. These authors also reported that fucoidan treatment resulted in restoring the normal expression of peroxisome proliferator-activated receptor gamma coactivator 1-alpha (PGC-1α) and nuclear transcription factor 2 (Nrf-2), thereby rescuing the mitochondrial functioning [14]. A study by Park et al. confirmed that pretreatment with fucoidan extracted from *E. cava* also showed antioxidant effects and protection of mitochondrial health in mice with cognitive dysfunction caused by trimethyltin injection [64]. It was also reported that fucoidan has a beneficial influence on inhibiting Bax expression and cytochrome c release, in accordance with previously reported in vitro findings [64]. (see also Figure 2) Finally, these same authors found significant decreases in Aβ formation and phosphorylation of Tau protein suggesting that fucoidan could be an efficacious agent for prevention for neurodegenerative disorders [64].

### Fucoidan, Neurodegeneration, and Sirtuin 3

As mentioned above, fucoidan may exert its beneficial effects via SIRT3 (as seen with TBI), and it appears that SIRT3 commonly presents a component of neurodegenerative diseases, as for example discussed by Meng et al. [56]. The most common neurodegenerative diseases are AD, PD, and Huntington Disease (HD). In the wake of several recent failures of AD therapies targeting beta-amyloid in plaques, growing evidence has suggested that infection with the herpes simplex virus (e.g., (HSV)-1) may play a role in AD. It is known that HSV-1 induces formation of beta-amyloid, and abnormally phosphorylated, AD-like tau (P-tau), which are the characteristic abnormal molecules of AD brains [65]. Furthermore, it has been found that SIRT3 may be a relevant therapeutic target in ALS. This suggests that SIRT3 may present a target for treatment of various neurodegenerative disorders [66]. Neurons have high energy demands, and dysregulation of mitochondrial quality and function is an important cause of neuronal degeneration (See also Figure 1). Meng et al. discussed that the *mitochondrial deacetylase* SIRT3 has been found to have a large effect on mitochondrial function [56]. Recent studies have also shown that SIRT3 has a role in mitochondrial quality control, including the refolding or degradation of misfolded/unfolded proteins, mitochondrial dynamics, mitophagy, and mitochondrial biogenesis, all of which are part and parcel of neurodegenerative diseases. Thus, the finding that fucoidan appears to interact with SIRT3, as seen with neurodegeneration due to TBI, may be very relevant

To recapitulate a bit, sirtuins are highly conserved NAD+ dependent class III histone deacetylases and catalyze deacetylation and ADP ribosylation of a number of non-histone proteins [67]. In the recent past, clusters of protein substrates for SIRT3 were identified in mitochondria and are now considered to be in association with protection from stress induced mitochondrial integrity and energy metabolism (Figure 1). In this way, SIRT3 may be protective regarding the pathogenesis of almost all neurodegenerative diseases. Some recent findings demonstrated that SIRT3 overexpression could prevent neuronal derangements in certain in vivo and in vitro models of aging and neurodegenerative brain disorders, including AD, PD, HD, TBI, stroke, etc. Similarly, loss of SIRT3 has been found to accelerate neurodegeneration in the brain challenged with excitotoxicity, which may explain the increase of SIRT3 levels found in the Wang’s et al. study [51] of TBI and its treatment with fucoidan, described above.

## 7. Fucoidan is Anti-inflammatory

Fucoidan treatment of meningitis in rats reduced all inflammatory changes, while fucoidan treatment of animals without meningitis increased blood white cell count [68]. This study also validated that selectins are involved in the early phase of pneumococcal meningitis and, possibly, are a target for adjunctive therapy with fucoidan [68]. In a rabbit meningitis model based on intracisternal injection of live *Streptococcus pneumoniae*, inhibition of leukocyte rolling by i.v. application of the polysaccharide fucoidan prevented the enhanced leukocyte extravasation into the subarachnoid space (SAS) and also attenuated the leakage of plasma proteins over the BBB [69]. Thus, fucoidan’s ability to block leukocyte rolling via binding to P-selectin presents the potential to reduce leukocyte-dependent CNS damage in bacterial meningitis. In addition, in C6 glioma cells it was found that fucoidan can suppress TNF-α and IFN-ɣ-induced NO production and iNOS expression, another way whereby fucoidan can attenuate inflammatory responses (See Figure 2). (Furthermore, fucoidan can inhibit: (1) TNF-α and IFN-ɣ-induced AP-1, IRF-1, JAK/STAT activation; (2) p38 mitogen-activated protein kinase (MAPK) activation; and (3) induced scavenger receptor B1 (SR-B1) expression [70]. In support, in vitro experiments with primary microglia indicated that the excessive production of TNF-α and ROS in LPS-induced primary microglia was significantly inhibited by fucoidan administration [71]. Finally, in animal studies, it was found in LPS treated rats that fucoidan significantly improved the behavioral functioning, prevented the loss of dopaminergic neurons, and inhibited the deleterious activation of microglia in the substantia nigra pars compacta [71]. These studies indicate the anti-inflammatory properties of fucoidan at cellular as well as systemic levels.

## 8. Fucoidan and Brain Infections (Prion and Virus)

Creutzfeldt-Jakob disease is a serious and lethal brain damaging condition. It has been demonstrated that sulfated glycans such as fucoidan and pentosan polysulfate, as well as amyloidophilic compounds such styrylbenzoazole derivatives, and phenylhydrazine derivatives present efficacies in prion-infected animals [72]. Wozniak et al. moved on to investigate the antiviral activity of sulfated fucans from five brown algae (*Scytothamnus australis*, *Marginariella boryana*, *Papenfussiella lutea*, *Splachnidium rugosum*, and *Undaria pinnatifida*) in relation to the HSV1-induced formation of beta-amyloid, and AD-like tau [65]. Four sulfated fucan extracts each prevented the accumulation of HSV1-induced beta-amyloid and AD-like tau in HSV1-infected Vero cells [65]. Thus, knowledge regarding fucoidan as an antiviral agent, beyond anti-inflammatory effects, may be relevant for brain disorders.

## 9. Fucoidan as Antiviral Agent

As the burden of viral infections is increasing, culminating with highly contagious new corona viruses displaying heterogeneous structures, complicating the design of targeted therapy, it is valuable to consider novel and safe antiviral agents [73,74]. In light of these notions, we assume that together with its antiviral and anti-inflammatory properties, fucoidan may deliver protective effects against many illnesses, including neurodegenerative and hepatic disorders, as reported and reviewed here and elsewhere. These effects of polysaccharides are mostly achieved by stimulating the production of viral antibodies and upregulation of interleukins (particularly IL-1 and IL-2), hence increasing the activation of macrophages and natural killer (NK) cells and promoting phagocytosis. In support of the above, it is assumed that the sulfate groups in fucoidan molecule may also contribute to the antiviral activity by acting as polyanions, and inhibiting cell surface interactions of positively charged viral domains with the host cells, thus preventing their penetration and/or adsorption. Boosting the humoral immune response by increased immunoglobulin synthesis has also been acknowledged as one of the antiviral effects of polysaccharides [74]. Early in vitro reports suggested that algal polysaccharides successfully inhibited the herpes simplex virus (HSV1 and HSV2), human cytomegalovirus, and bovine viral diarrhea virus [75,76], which may also have implications for brain diseases, including Alzheimer [65]. A study conducted by Hidari et al. placed the dengue virus type 2 (DEN 2), in the list of successfully inhibited viral species by fucoidan extracted from *C. okamurans* in the BHK-21 cell line, additionally defining glucuronic acid and the sulfated fucose contents as key elements involved in antiviral activity of this macromolecule [77]. These authors also established the significance of arginine-323 as a vital region in the envelope glycoprotein (EPG), which enables the interactions of glucuronic acid with the virus, while its possible substitution or “mispositioning” may decrease the antiviral effects of fucoidan as shown for DEN-1 and DEN-3 viruses [77]. Conversion of the complex structure of polysaccharides into monosaccharides could also diminish their virucidal activity [78].

From recent published results, Sun et al. verified these findings by establishing the inhibitory activities of two low molecular fractions of fucoidan isolated from *L. japonica* against influenza A, adenovirus, and parainfluenza virus type 1 (HPIV1) in Hep-2, Hela, and MDCK cells [79]. These authors also reported that acute intraperitoneal treatment with these fucoidan fractions prolonged the average survival time and increased the viability of lung, thymus, and spleen cells [79]. Similar research conducted by Wang et al. revealed the inhibition of *neuraminidase* and epidermal growth factor cellular pathway (EGFR) of fucoidan as crucial molecular mechanisms for preventing the penetration of influenza H1N1 virus into host cells [80]. Another study showed the inhibitory properties of crude fucoidans obtained from the brown algae *D. bartayesiana* and *T. decurrens* of the human immunodeficiency virus (HIV) in infected peripheral blood mononuclear cells (PBMCs) at the extremely low inhibitory concentrations of 1.56 µg/mL and 3 µg/mL, respectively, thus exhibiting significantly higher antiviral effects than ribavirin [78,81]. Prokofjeva et al. also acknowledged fucoidans as potent anti-HIV agents irrespective of their degree of sulfation and carbohydrate structure [82].

Since fucoidan has been proven quite successful in inhibiting single stranded RNA (ssRNA) respiratory viruses such as influenza A and HPIV1, in addition to virucidal activity against DEN (positive ssRNA virus), we are also led to believe that it could be influential in treating coronaviruses induce diseases such as COVID-19, given COVID-19’s similarities with the genetic material and symptomatology of the virus species mentioned here [83]. The variety of the cellular mechanisms by which this polysaccharide achieves its antiviral effects probably contributes to its potency.

## 10. Fucoidan and Brain (Excito)toxicity

Fucoidan was shown to suppress increased oxidative stress in bovine brain microvessel endothelial cells (BBMECs) in culture after exposure to diesel exhaust particles (DEPs) [84]. In addition, permeability of BBMECs induced by DEP exposure was decreased by fucoidan treatment. This study provides evidence that fucoidan might protect the CNS against toxic effects of DEP exposure [84]. In cultured cortical neurons from one-day old Wistar rats, fucoidan suppressed NMDA induced Ca^2+^ responses by 100% [85]. However, the Ca^2+^ responses of hippocampal neurons induced by glutamate, ACPD, or adrenaline showed only slight decreases following fucoidan treatment. Nonetheless, in cortical as well as hippocampal neurons, fucoidan treatment significantly decreased mRNA expression of the NMDA-NR1 receptor and the primer pair for l-type Ca^2+^ channels, namely, PR1/PR2. In this way, fucoidan may counteract excitotoxicity, at least in the cerebral cortex [85].

Previous work showed that the glycosaminoglycan (GAG) dextran sulfate (500 kDa) altered the binding and channel properties of alpha-amino-3-hydroxy-5-methyl-4-isoxazolepropionic acid (AMPA)-type glutamate receptors [86]. Dextran sulfate was more potent in inhibiting high-affinity AMPA binding to solubilized receptors (EC (50) of 7 nM) than fucoidan, another GAG (EC(50) of 124 nM). Additionally, dextran sulfate (2 nM), produced a three- to four-fold increase in open channel probability and a three-fold increase in mean burst duration of channel activity elicited by 283 nM AMPA. Fucoidan produced similar effects, but at a concentration several times higher than that of dextran sulfate [86]. These findings suggest that GAG components of proteoglycans can interact with and alter the binding affinity of AMPA receptors and modulate their functional properties. Thus, in addition to NMDA receptors, fucoidan can also prevent excitotoxity by altering the affinity of AMPA receptors. We hope that these fucoidan effects can be optimized, for example, by producing more efficacious fucoidan derivatives.

## 11. Fucoidan and Alzheimer Disease (AD)

As mentioned above, fucoidan can counteract AD-like adverse effects of HSV1 infection*,* such as the formation of beta-amyloid and abnormal P-tau. Other studies showed with rat behavioral tests that fucoidan can ameliorate Aβ (1-40)-induced learning and memory impairments [87]. Furthermore, fucoidan reversed the decreased activity of choline acetyl transferase (ChAT), superoxide dismutase (SOD), glutathione peroxidase (GSH-Px), and acetylcholine (Ach) content, as well as the inhibitory effects on malondialdehyde (MDA) synthesis in hippocampal tissue of Aβ-injected rats [87]. Moreover, these effects were accompanied by an increase of Bcl-2/Bax ratio and a decrease of caspase-3 activity [87]. Thus, apparently, by regulating the cholinergic system, reducing oxidative stress, and inhibiting apoptosis, fucoidan can ameliorate Aβ-induced AD.

Park et al. investigated the sea weed *Ecklonia cava* (*E. cava*) for the effects of fucoidan extract on cognitive function [64]. They applied Y-maze, passive avoidance, and a Morris water maze to a trimethyltin (TMT)-induced cognitive dysfunction model. This demonstrated that the fucoidan extract promoted learning and memory improvements. In mouse brain tissue taken after such behavioral tests, fucoidan extract was shown to provide inhibitory effects on lipid peroxidation and improvement of cholinergic system activity. Mitochondrial activity was improved as seen from associated mitochondrial ROS content and mitochondrial membrane potential (MMP, ΔΨm) levels, and was also detected by mitochondria-mediated protein (BAX, cytochrome C) analysis for apoptosis induction. It appeared that the fucoidan-rich substances from *E. cava* could improve cognitive functions by downregulating amyloid-β production and tau hyperphosphorylation [64].

Jhamandas et al. showed that fucoidan reduced cell death rates otherwise induced by A beta (25-35) or A beta (1-42) to rat cholinergic basal forebrain cultures [88]. In this study, it was also found that fucoidan attenuated A beta-induced downregulation of phosphorylated protein kinase C. Furthermore, A beta (1-42)-induced generation of ROS was blocked by prior exposure of the cultures to fucoidan. Regarding apoptosis, A beta activation of caspases 9 and 3 is blocked by pretreatment of cultures with fucoidan [88]. Caspases 9 and 3 are well known components of the signaling pathways of apoptotic cell death induction. These results show that fucoidan has neuroprotective effects against A beta-induced neurotoxicity in basal forebrain neuronal cultures, which may have implications for AD and other neurodegenerative diseases

### Sirtuin and Alzheimer Disease (AD)

As fucoidan can affect SIRT3 function, it is interesting to know what SIRT3 may do in AD. Lee et al. have discussed mitochondrial dysfunction in connection with the pathogenesis of AD [89]. In particular, SIRT3 mRNA and protein levels are significantly decreased in AD cerebral cortex, and Ac-p53 K320 is significantly increased in AD mitochondria. In this context, SIRT3 prevented p53-induced mitochondrial dysfunction and neuronal damage in a deacetylase activity-dependent manner. Notably, mitochondrial targeted p53 (mito-p53), directly reduced mitochondria DNA-encoded ND2 and ND4 gene expression, resulting in increased ROS and reduced mitochondrial oxygen consumption. Interestingly, ND2 and ND4 gene expressions are significantly decreased in patients with AD and increased p53 occupancy in mitochondrial DNA in AD. Lee et al. further found that SIRT3 overexpression restored the expression of ND2 and ND4 and improved mitochondrial oxygen consumption by repressing mito-p53 activity [89]. These results indicate that SIRT3 dysfunction leads to p53-mediated mitochondrial and neuronal damage in AD. Therapeutic modulation of SIRT3 activity may ameliorate mitochondrial pathology and neurodegeneration in AD. In the light that fucoidan can modulate SIRT3 activity, it emphasizes that fucoidan may present an agent to ameliorate AD.

## 12. Fucoidan and Parkinson Disease (PD)

The effects of fucoidan has been studied intensively on several PD models, for example: (i) in cell culture of dopaminergic nerve cells; (ii) MPTP (1-methyl-4-phenyl-1,2,3,6-tetrahydropyridine) application to mice; and (iii) application of 6-hydroxydopamine and rotenone application to rats, as given in some detail below.

In a study by Liang et al. on a cultured dopaminergic nerve precursor cell line (MN9D), cell viability decreased by 50% within 24 h of 100 μM MPP+ application [13]. 1-methyl-4-phenylpyridinium (MPP+), the toxic bioactivation product of MPTP, is a toxic compound that via the dopamine transporter is brought into neurons, in which it initiates neuronal death by inhibiting complex I of the mitochondria. Pretreatment with 100 μM fucoidan in this paradigm reversed the reduction of SOD and GSH, as well as the decreased cell viability and induced apoptotic cell death, otherwise brought about within 6 h by MPP+. Furthermore, preceding this fucoidan reduced cellular expression of LC3-II and CatD within 3 h and suppressed the induction of Bax protein. Thus, Liang et al. suggested that fucoidan may have a positive, curative effect regarding PD [13].

Regarding animals, Luo et al. applied MPTP to C57/BL mice [90]. When fucoidan was administered prior to MPTP, behavioral deficits were reduced, and levels of striatal dopamine and its metabolites were enhanced, cell death was reduced, and a marked increase in tyrosine hydroxylase expression relative to mice treated with MPTP alone was also observed. Furthermore, as in the Liang et al. study [13], it was found that fucoidan inhibited MPTP-induced lipid peroxidation and reduction of antioxidant enzyme activity. In addition, pre-treatment with fucoidan significantly protected against MPP(+)-induced damage in MN9D cells [90].

In a 6-hydroxydopamine (6-OHDA) rat model of PD, chronic fucoidan administration mitigated the motor dysfunction otherwise induced by 6-OHDA [91]. Similarly, fucoidan reduced the loss of DA neurons in the SNc and DA fibers in the striatum in 6-OHDA-lesioned rats. Moreover, fucoidan inhibited the 6-OHDA-stimulating expression of Nox1 in both tyrosine hydroxylase (TH)-positive neurons as well as non-TH-positive neurons, and prevented Nox1-sensitive oxidative stress and cell damage in SNc neurons. Finally, fucoidan also effectively inhibited nigral microglial activation [91].

In a rotenone-induced PD rat model, it was found that chronic treatment with fucoidan significantly reversed the loss of nigral dopaminergic neurons, striatal dopaminergic fibers, and reduction of striatal dopamine levels [14]. Fucoidan also alleviated rotenone-induced behavioral deficits. Interestingly, in the substantia nigra of these PD rats, the reduced mitochondrial respiratory function, detected by the mitochondrial oxygen consumption and the expression of peroxisome proliferator-activated receptor gamma coactivator 1-alpha (PGC-1α) and nuclear transcription factor 2 (NRF2), was markedly reversed by fucoidan [14]. Furthermore, oxidative products induced by rotenone were significantly reduced by fucoidan. These results thus also suggest some functional pathways modulated by fucoidan in relation to the neurodegenerative disease of PD.

Taken together, these studies indicate that fucoidan attenuates PD characteristics induced in various animal and cell culture models for PD. Thus, fucoidan provides a promising venue for treatment of PD by modulating its various underlying molecular biological mechanisms.

Sirtuin and Parkinson Disease (PD)

It may be that as in TBI, and also suggested for AD, fucoidan exerts its ameliorating effects regarding PD via interactions with SIRT3. Interestingly, while it was shown that SIRT3 null mice do not exhibit motor and non-moto deficits compared with wild-type controls, SIRT3 deficiency dramatically exacerbated the degeneration of nigrostriatal dopaminergic neurons in 1-methyl-4-phenyl-1,2,3,6-tetrahydropyridine (MPTP)-induced PD mice [92]. SIRT3 null mice exposed to MPTP also exhibited decreased SOD 2, a specific mitochondrial antioxidant enzyme, and reduced glutathione peroxidase expression compared with wild-type controls [92]. Taken together, these findings strongly support that SIRT3 has a possible role in MPTP-induced neurodegeneration via preserving free radical scavenging capacity in mitochondria. Thus, it would be worthwhile to study fucoidan-SIRT3 interactions in models for PD. It indeed is very tempting to assume that fucoidan also interacts with SIRT3 in PD.

## 13. Fucoidan and Stroke

Stroke is one of the leading causes of death. Growing evidence indicates that ketone bodies have beneficial effects in treating stroke, but their underlying mechanism remains unclear [93].

The potential of fucoidan to ameliorate stroke injury in the brain has been of interest for more than two decades now. Because intracerebral hemorrhage (as induced by injection of bacterial collagenase into the caudate nucleus) is associated with more inflammation than seen with ischemic stroke, this stroke model attracted early interest for fucoidan treatment testing, as fucoidan can counteract inflammatory responses [94]. Fucoidan-treated rats exhibited evidence of impaired blood clotting and hemodilution, had larger hematomas, and tended to have less inflammation in the vicinity of the hematoma after three days. Interestingly, fucoidan-treated rats showed significantly more rapid improvement of motor function in the first week following hemorrhage and better memory retention in the passive avoidance test, as compared to untreated controls [94]. This early study from 1999 stated that investigations of more specific anti-inflammatory agents and hemodiluting agents would be warranted in intracerebral hemorrhage [94].

As it was also understood that leukocyte-endothelial adhesion is a key step to initiate post-ischemic reperfusion injury in many organs, this potential contribution to stroke, including fucoidan interference, was also studied. Uhm et al. found that the expressions of P-selectin mRNA and protein were increased in the ipsilateral hemisphere with a peak at 8 h after hypoxia-ischemia in immature brain [95]. Such temporal profiles of P-selectin expression followed by hypoxia-ischemia are consistent with a role in the subsequent brain injury. Because fucoidan is known to inhibit P/L-selectin mediated leukocyte adhesion, it was examined whether the treatment of fucoidan attenuates hypoxia-ischemia-induced neural damages [95]. Indeed, fucoidan presented a substantial neuroprotective effect, including significant inhibition of the leukocyte adhesion, as revealed by myeloperoxidase activity. These results suggest that anti-adhesion strategy as can be provided by fucoidan may be an effective therapeutic application for perinatal hypoxic-ischemic encephalopathy [95].

The following study presented further interest in fucoidan as an anti-inflammatory agent. Fucoidan treatment inhibited the expressions of some brain cytokine or chemokine mRNA, such as IL-8, TNF-α, and iNOS in the brain of the rats treated only with LPS [96]. Moreover, fucoidan treatment dramatically decreased the infarct size in accelerated cerebral ischemic injury induced by LPS treatment. In addition, the immunoreactivity of myleoperoxidase (MPO), a marker for quantifying neutrophil accumulation, was distinctively decreased in the ischemic brain of the fucoidan-treated rat. In brief, the results of Kang et al. [96] indicated that fucoidan provides a neuroprotective effect on LPS accelerated cerebral ischemic injury through inhibiting the expression of some cytokine/chemokine and neutrophil recruitments [96].

Intracerebral hemorrhage (ICH) is the most fatal stroke subtype, with no effective therapies [97]. Additionally, fucoidan did not have effects on brain water content, neurological deficits, and hemoglobin content after ICH. The authors suggested that this may be so because crude fucoidan was used in this study, and high-molecular-weight fucoidans (HMWF) are reported to have less therapeutic potential than LMWF’s [97].

The effects of fucoidan on cerebral ischemia-reperfusion injury (IRI) including the inflammatory and other underlying mechanisms were further explored in Sprague-Dawley (SD) rats [98]. The results showed that administration of fucoidan significantly reduced the neurological deficits and infarct volume in a dose-dependent manner. Additionally, fucoidan significantly decreased the levels of: (i) inflammation-associated cytokines (interleukin (IL)-1β, IL-6, myeloperoxidase (MPO), and tumor necrosis factor (TNF)-α); (ii) oxidative stress-related proteins malondialdehyde (MDA) and SOD; (iii) apoptosis (in particular, apoptosis-related proteins (p-53, Bax, and B-cell lymphoma (Bcl)-2)); and (iv) the MAPK pathway mitogen-activated protein kinase (MAPK) pathway (in particular, phosphorylation-extracellular signal regulated kinase (p-ERK), p-c-Jun N-terminal kinase (JNK), and p-p38). Thus, fucoidan’s protective role in cerebral IRI includes anti-inflammatory effects, anti-apoptotic effects, anti-oxidative stress affects, and potentially gene expression regulation [98].

Studies directed at better determining the stroke-fucoidan interaction in various brain cell types, at systemic, in situ levels, showed that pretreatment with fucoidan confers neuroprotection against transient global cerebral ischemic injury in the gerbil hippocampal CA1 area via reducing of glial cell activation and oxidative stress [99]. In some detail, the neuroprotective effect of fucoidan against transient global cerebral ischemia (tGCI) included inhibition of activation of astrocytes and microglia in the ischemic CA1 area. Furthermore, it significantly reduced otherwise increased 4-hydroxy-2-noneal and superoxide anion radical production in the ischemic CA1 area with subsequent increased expressions of SOD1 and SOD2 in the CA1 pyramidal neurons before and after tGCI [99]. Additionally, in obese gerbils, fucoidan treatment attenuated acceleration and exacerbation of tGCI-induced neuronal death in the CA1-3 hippocampal areas, and levels of oxidative stress indicators (dihydroethidium, 8-hydroxyguanine, and 4-hydroxy-2-nonenal) were significantly reduced, while levels of antioxidant enzymes (SOD1 and SOD2) were significantly increased in pre- and post-ischemic phases [100]. These findings indicate that pretreatment with fucoidan can relieve the acceleration and exacerbation of ischemic brain injury in an obese state via the attenuation of obesity-induced severe oxidative damage, and related factors.

### Sirtuin and Stroke

As mentioned at a few occasions above, SIRTs are a family of NAD+ dependent histone deacetylase (HDAC) proteins implicated in aging, cell cycle regulation, and metabolism. These proteins are involved in the epigenetic modification of neuromodulatory proteins after stroke via acetylation/deacetylation. The specific role of SIRT3, a mitochondrial sirtuin, in post-stroke injury has been relatively unexplored. Nonetheless, Verma et al. [101] showed that SIRT3 knockout (KO) mice show significant neuroprotection at 3 days after ischemia/reperfusion (I/R) or stroke injury. The deacetylation activity of SIRT3, measured as the amount of reduced acetylated lysine, was increased after stroke [101]. In male SIRT3 KO mice and wild-type littermates (WT), stroke-induced increases in liver kinase 1 (LKB1) activity were also appeared reduced in KO mice at 3 days after stroke.

Yin et al. investigated whether mitochondrial SIRT3 could mediate the neuroprotective effects of ketone bodies after ischemic stroke. The ketone treatment did enhance mitochondrial function, reduced oxidative stress, and probably in this way reduced infarct volume. This was associated with improved neurologic function after ischemia, including the neurologic score, the performance in rotarod, and in open field tests. They further presented that ketones’ effects were achieved by upregulating SIRT3 and its downstream substrates forehead box O3a (FoxO3a) and superoxide dismutase 2 (SOD2) in the penumbra region. This appeared likely, since knocking down SIRT3 in vitro diminished ketones’ beneficial effects [93]. It also indicates that upregulation of SIRT3 after stroke is beneficial for amelioration of brain damage caused by stroke. These results provide us a foundation to develop novel therapeutics targeting this SIRT3-FoxO3a-SOD2 pathway. Of course, in the framework of this review, we would like to suggest the potential of fucoidan, based on its known modes of action and its efficaciousness.

On further investigation, Verma et al. found that the levels of SIRT1, another important member of the Sirtuin family, were increased in the brains of Sirt3 KO mice after stroke [101]. To determine the translational relevance of these findings, they tested the effects of pharmacological inhibition of SIRT3. They found no benefit of SIRT3 inhibition despite clear evidence of deacetylation. Overall, it was concluded that SIRT3 KO mice show neuroprotection by a compensatory rise in SIRT1 rather than the loss of SIRT3 after stroke [101]. Further analysis will determine the importance of using both pharmacological and genetic methods in pre-clinical stroke studies to better understand molecular biological pathways [101]. Additionally, it is important to further investigate the interaction between SIRT1 and SIRT3.

More studies approached the question of SIRT3 involvement in brain damage due to stroke. The role of SIRT3 in microglial cell migration and invading macrophages in ischemic stroke was studied. The middle cerebral artery occlusion (MCAO) animal model of focal ischemia was used [102]. Lentivirus-packaged SIRT3 overexpression was applied, and also knock down in microglial N9 cells in culture, to investigate the underlying mechanism driving microglial cell migration. More microglial cells appeared in the ischemic lesion side after MCAO. The levels of SIRT3 were increased in macrophages invading after ischemia. CX3CR1 levels were increased with SIRT3 overexpression. Furthermore, SIRT3 promoted microglial N9 cells migration by upregulating CX3CR1 in both normal and glucose deprived culture media [102]. These effects were G protein-dependent. Thus, SIRT3 promotes microglia migration by upregulating CX3CR1. This appears counter intuitive, finding that SIRT3 promotes microglia migration, while the consensus appears to be that SIRT3 acts anti-inflammatory. However, as mentioned, it would be nice also to further investigate interaction between SIRT1 and SIRT3.

After transient middle cerebral artery occlusion (tMCAO) in adult male SIRT3 KO and wild-type (WT) mice, it was found that the level of SIRT3 in infarct region is decreased after ischemic stroke [103]. In addition, it was found that SIRT3 KO mice showed worse neurobehavioral outcome compared with WT mice, accompanied by decreased neurogenesis and angiogenesis, as shown by the reduction in number of DCX+/BrdU+ cells, NeuN+/BrdU+ cells, and CD31+/BrdU+ cells in the perifocal region during the recovery phase after ischemic stroke [103]. Furthermore, SIRT3 deficiency reduced the activation of vascular endothelial growth factor (VEGF), AKT, and extracellular signal-regulated kinases (ERK) signaling pathways [103]. These results indicate that SIRT3 is beneficial to neurovascular and functional recovery following chronic ischemic stroke. As a concluding remark, as TBI and stroke are brain injuries, it could be very fruitful to investigate fucoidan–sirtuin interactions in stroke models, and see whether similarities with TBI models will be seen.

## 14. Fucoidan and Liver Injury

Regarding the previously mentioned conclusions about antioxidant, anti-inflammatory, and anti-cell death impact of fucoidan, it can be deduced that this polysaccharide could have a significant therapeutic effect against systemic and whole organism disorders and inflammations rather than just impairments of tissues. It could also have a significant impact on regenerating the antioxidant potentials of more dynamic cells with significantly higher levels of ROS production such as hepatocytes [104]. We will discuss fucoidan-liver interactions here. An additional point, not to be dwelt on, may be the effect of liver damage on brain health, as for example with hepatic encephalopathy, or maybe also atherosclerosis associated with impaired liver functions. Importantly, restricting ourselves to the liver, hepatocytes are easily prone to structural alterations, cellular deterioration, and damage, given their rates of metabolic activity; thus they are ideal models for confirming the protective effects of fucoidan [104]. Indeed, fucoidan has shown cytoprotective effects against the hepatotoxicity of several xenobiotics, such as acetaminophen (APAP) [105] or carbon tetrachloride (CCl_4_) [106] (Figure 3). Li et al. also confirmed the effects of fucoidan, extracted from *F. vesiculosus,* against liver fibrosis induced by acute CCl_4_ treatment (Figure 3) or bile duct ligation (BDL) through decreasing serum transaminase activity and hydroxyproline concentration; however, relative large doses of fucoidan were needed to restore normal levels of serum transaminase activity and hydroxyproline concentration [15] (see also Figure 3). Moreover, they discovered that fucoidan significantly reduced synthesis of collagen type 1 and alpha smooth muscle actin (α-SMA) proteins, which are typically upregulated in hepatocyte injury stimulating liver fibrosis by transformation of hepatic stellate cells (HSCs) to myofibroblasts [107]. Furthermore, the authors proved the inhibitory effect of fucoidan on transforming growth factor beta (TGF-β)/Smad molecular pathways, by the decreased expression of Beclin-1, a transcription factor activated by this pathway which influences autophagosome occurrence, which was considerably increased in the CCl_4_ or BDL injured hepatocytes (Figure 3) [15,108]. A recent study confirmed these findings with fucoidan extracted from *S. fluitans* on Wistar rats [109]. Thus, it can be concluded that fucoidan exerts its hepatoprotective effects through the alteration of pathways directly included in modification of liver microenvironment.

The contribution of sulfate content for the beneficial effects of this algal polysaccharide on hepatic injury has also been studied. Namely, Liu et al. reported more significantly decreased levels of serum lactate dehydrogenase (LDH) levels in mice with acute CCl_4_ treatment after co-treatment with more sulphated fractions of fucoidan (24.65% and 29.31%) isolated from *K. crassifolia* [110] (see also Figure 3). However, the authors did not find major differences in the inhibition of lipid peroxidation and GSH restoration in animals treated with different doses of fucoidan fractions [110]. Similar results were also found for diminishing AST and MDA levels during treatment with high sulphated fraction of fucoidan (27.08%) from *S. japonica*, supported by histopathological analyses, which revealed a complete inhibition of liver necrosis by this fraction of fucoidan, with a mega dose applied [111]. Findings about the dependence of sulphate content on the biological potency of algal polysaccharides are in corroboration with those published by Wang et al., who reported that low sulphate fraction (1%) of fucoidan from *C. costata* did not achieve suppression of alanine aminotransferase (ALT) and aspartate aminotransferase (AST) levels [112].

Recent investigations about fucoidan hepatoprotective effects against drug induced liver injury also provided promising results. A study by Abdel-Daim et al. has shown that acute treatment of rats with high quantity of fucoidan efficiently restores transaminase levels, creatine kinase, and acetylcholinesterase activity, as well as cholesterol and creatinine serum concentrations, after diazinon-induced hepatic injury, together with the dose dependent decrease of inflammatory biomarkers, such as TNF-α and IL-6 [113]. Similarly, neurons, hepatocytes, renal cells, and cardiomyocytes also exhibited improvement of the enzymatic antioxidants via the restoration of cellular Cat, SOD, and GPx activities and a significant decrease in lipid peroxidation and nitric oxide levels (Figure 3) [113]. The same results were obtained using mice as animal models for microcystin- LR injury [114]. Discoveries of Wang et al. suggested that fucoidan exerts its protective effects against APAP liver injury mostly through activating the Nrf2-ARE molecular pathway in addition to suppressing cytochrome 450 member CYP2E1, responsible for metabolizing acetaminophen to its toxic product N-acetyl-p-benzoquinoneimine (NAPQI) and triggering GSH depletion [16,115] (see also Figure 3). However, large doses of fucoidan were able to increase the antioxidant response of liver cells and suppress serum ALT, AST, and LDH levels only at the early stages of injury (up to 2 h after of APAP injection) [16] (see also Figure 3).

### 14.1. Sirtuin and Liver Injury

Reminiscent of neurons, fucoidan also stimulates expression of sirtuin molecules (SIRT1) in hepatocytes, mostly involved in regulation of glucose and lipid metabolism in the liver, thus being one of the crucial factors included in the pathophysiology of the metabolic syndrome (MetS) and insulin resistance [116]. Considering the severity of these illnesses and the increasing trend of obesity worldwide, it is of great importance to discover safe therapeutics that would decrease these alarming rates [117]. In this context, a study by Zheng et al. reported significant reduction of plasma and liver cholesterol and triglycerides in db/db mice after sub-chronic treatment with low molecular fucoidan isolated from *L. japonica*, but failed to acknowledge reduction in fasting glucose levels [118]. Hepatoprotective effects were confirmed by inhibition of transaminase release, but only at the highest dose applied. As expected, the anti-inflammatory and antioxidant effects of fucoidan in vivo were underlined by the significant decreases in the cytokines and ROS markers, which was confirmed by the reduced expression of NF-κB in the fucoidan treated mice [118]. Most importantly, the authors reported significantly elevated levels of SIRT1 and 5’ adenosine monophosphate-activated protein kinase (AMPK), whose activation is typically associated with increased glucose uptake, fatty acid oxidation, and glycolysis, which certainly explains these antilipotoxic effects of fucoidan (Figure 3) [118]. Treatment with *F. vesiculosus* fucoidan on palmitate induced insulin resistant HepG2 cells also resulted in increased expression of AMPK, additionally confirmed by their increased glucose consumption and decreased lipid profile [119]. Furthermore, sub-chronic administration of fucoidan isolated from the same species also affected Low-Density Lipoprotein Cholesterol (LDL-C) and High-Density Lipoprotein Cholesterol (LDL-C). In particular, this fucoidan application also resulted in reduction of LDL-C and elevation of HDL-C levels in high fat diet fed mice, thus showing similar results with metformin in most of the analyzed parameters both in vivo and in vitro (except for HDL-C), including body weight decreases, which additionally promotes fucoidan as an efficient remedy for metabolic disorders, considering the ongoing controversies of the effects of metformin [119,120]. Another study dealt with aspects of homeostasis, including glucose metabolism, influenced by fucoidans [121]. Briefly, the effects of low-molecular-weight fucoidan (LMWF) in terms of antihyperglycemic, antihyperlipidemic, and hepatoprotective activities, were investigated in a mouse model of type II diabetes. Blood sugar levels and increased serum adiponectin levels, were lowered by LMWF intake; thus they are relatively effective at improving hepatic glucose metabolism [121].

### 14.2. Liver Injury, Cholesterol, Atherosclerosis, and Fucoidan

From the findings about hypolipidemic effect of fucoidans, the possible suppression of atherosclerosis by this molecule can be derived from complex pathology of atherosclerosis, which combines dyslipidemia, inflammation, and atherothrombosis, mostly affected by increased levels of plasma LDL and its oxidative transformations as a trigger for occlusion of peripheral arteries [122]. In parallel, the explained hepatoprotective features of fucoidan also go in favor of this hypothesis considering the impacts of impaired hepatic functioning on lipid homeostasis, regulation of metabolic pathways and lipoprotein uptake [123]. The revealed contribution of ROS in the development of atherosclerotic plaques, as confirmed by increased lipid peroxidation, glutathione depletion, and plasma and tissue protein carbonylation levels, also endorsed investigations of fucoidan as suitable therapeutic for this disease [124,125,126]. A study by Park et al. reported the significant decreases in plasma lipid profiles in fucoidan-treated mice with chemically-induced hyperlipidemia, which were comparable with statin effects [127]. Furthermore, it was concluded that this marine derived polysaccharide influences significant reduction in the development of aortic lesions in chronic atherosclerosis model and reduced expression of lipogenic enzymes *fatty acid synthase* (FAS) and *acetyl CoA carboxylase* (ACC) in HepG2 cells [127]. This study also confirmed the significant changes in the expression of SREBP-2 and the LDL receptor, which makes them the probable molecular targets for hypolipidemic effects of fucoidan [127]. These findings were confirmed on apolipoprotein E deficient mice (ApoE^−/−^) in the study of Yin et al., using fucoidan isolated from *A. nodosum* [128]. Beside the amelioration of the lipid profiles and transaminase activity in the high fat diet treated mice, the sub-chronic treatment with fucoidan favored cellular cholesterol efflux by upregulation of ATP-binding cassette transporters (ABCA1 and ABCG8) and suppressed SREBP1 and peroxisome proliferator-activated receptor gamma (PPARγ), thereby inhibiting fatty degeneration in the liver. However, the treatment did not influence changes in LDL-R expression in ApoE^−/−^, in contrast with the effects of fucoidan isolated from *F. vesiculosus,* thus suggesting the disparity of the effects of polysaccharides with different structures as well as the chemical composition [128].

### 14.3. Summary, Liver and Fucoidan

To summarize, fucoidan averts hepatic injury and necrosis by: (i) inhibiting the profibrotic pathways in the extracellular matrix that promote HSCs production; (ii) modulating the cytochrome p450 enzyme activity and influencing the expression of Nrf-2 transcription factor, which stimulates the antioxidant response of hepatocytes; (iii) increasing the expression of SIRT1 and other molecules involved in regulation of lipoprotein metabolism. While fucoidan research is still in its early stages, even though it exhibits versatile molecular response in liver cells, thus far, research only proved that fucoidan can alleviates acute injury at relatively large doses. Thus, it would be worthwhile to go the road of optimizing fucoidan effects by enhancing the efficaciousness of its derivatives. Furthermore, possibly with its protective functions in the liver, fucoidan can suppress atherosclerosis, which of course would also present a beneficial factor regarding reduction of the incidence of stroke and explain some of its healing effects.

## 15. Differences in Physiological Activity of HMWF vs. LMWF, Contribution of the Sulfate Content

Another question arises: which type of fucoidan in general would be relatively advantageous to use—high molecular weight fucoidan (HMWF) or low molecular weight fucoidan (LMWF).

The bioactivity of fucoidan is primarily dependent on its molecular weight and sulfate content. In general, it is accepted that the therapeutic potentials of HMWF are limited due to its lower cellular uptake and bioavailability, as reported by several studies [9,129]. These limitations derive mainly from the difficulties of HMWF to cross lipid bilayers, thereby suggesting superior effects of LMWF regarding their anticancer and antioxidant activity, simply deriving from their relatively efficacious membrane permeability [74,130]. In one study, the structure/function relationships of fucoidans from *Ascophyllum nodosum* regarding their pro-angiogenic effect and cellular uptake in human endothelial cells were investigated [131]. It was evidenced that LMWF have relatively high pro-angiogenic and pro-migratory potential. This may be interesting knowledge for the potential application LMWF to vascular repair in ischemia. In contrast, HMWF seems to have greater immunostimulatory effects than LMWF in the spleen, as indicated by the increased activation of natural killer NK cells in addition to the higher levels of interleukins and TNF-α [132]. In accordance with the latter, a study of Liu et al. revealed that the HMWF exhibited more significant neuroprotective effect than the LMWF in SH-SY5Y cells, thereby suggesting that the amount of sulfate is an important factor for improving therapeutic properties of fucoidan [133]. This study also uncovered the advantageous efficient blockage of HMWF, which further reveals the complexity of its steric configuration providing more binding sites to the complementary factors [133].

In addition to distinctions in molecular weight also differences in presence of sulfated groups may contribute to the beneficial effects of fucoidans. Overall, a positive correlation exists between the sulfate content and antioxidant capability, which generally implies increased therapeutic impact of more sulfated fucoidans against diseases whose etiologies include oxidative damage [74]. More specifically, highly sulfated fucoidans have shown significant attenuation of lipid accumulation and antitumor activity [134,135]. In one review focusing on bone tissue regeneration, it was concluded that sulfated polysaccharides, including fucoidans, have exceptional properties in terms of hydrogel-forming ability, scaffold formation, mimicking the extracellular matrix, alkaline phosphatase activity, biomineralization ability, and stem cell differentiation [136].

## 16. Potential Mitochondrial Involvement in the Curative Effects of Fucoidans and Sirtuins

In a recent review by us, we discussed the potential involvement of a mitochondrial protein (the 18 kDa Translocator Protein; TSPO ) in brain disorders [137]. It is acknowledged that TSPO can serve to maintain homeostasis at organism, tissue, and cellular levels, including curative effects. Thus, in general, it is interesting to further investigate the potential of targeting mitochondria for curative effects on various aspects of brain disorders, including liver issues. As discussed throughout this review (indicated in Figure 1 and Figure 3), fucoidans via their interactions with Sirtuins 1 and 3, can affect mitochondrial functions. This includes homeostasis and metabolism, which are essential components for maintaining brain and liver health, including curative responses to injury and disease. In this respect, a study of Nogueiras et al. [138] recognized that by deacetylating a variety of proteins that induce catabolic processes, SIRT1 and SIRT3 coordinately increase cellular energy stores and ultimately maintain cellular energy homeostasis. It is also known that effects in the pathways controlled by Sirtuins 1 and 3 can result in various metabolic disorders [138]. Thus, our study suggests that studying the interactions of fucoidans with sirtuins can elicit multiple metabolic benefits regarding various forms of brain disorders and liver injuries.

## 17. Concluding Remarks

Fucoidan presents beneficial effects in brain and liver damage, due to injury and disease. An interesting consideration is that it is possible to modify fucoidan derivatives to modulate fucoidan effects. Additionally, it appears that fucoidan can interact with sirtuins; in the brain (SIRT3), this appears to be associated with mitochondrial function and modulation cell nuclear gene expression. In the liver (SIRT1), this appears to be associated with the regulation of glucose and lipid metabolism. Finally, in the brain, in particular the BBB, fucoidan interacts with P-selectin, thereby blocking macrophages from crossing and thus attenuating the inflammatory response in the brain proper. In this context, to emphasize here, it is becoming more and more recognized that prion, viral, and bacterial infections can induce neurodegeneration, as for example observed with Alzheimer Disease (AD) [139]. Thus, fucoidans’ anti-prion, anti-viral, and anti-bacterial functions may become relevant in this respect. We believe that since fucoidans have demonstrable curative effects on various brain disorders (and also other diseases not discussed in this review) it would be worthwhile to deepen research of the various effects of fucoidans at molecular and cellular levels and the whole organism in general.

## Figures and Tables

**Figure 1 marinedrugs-18-00242-f001:**
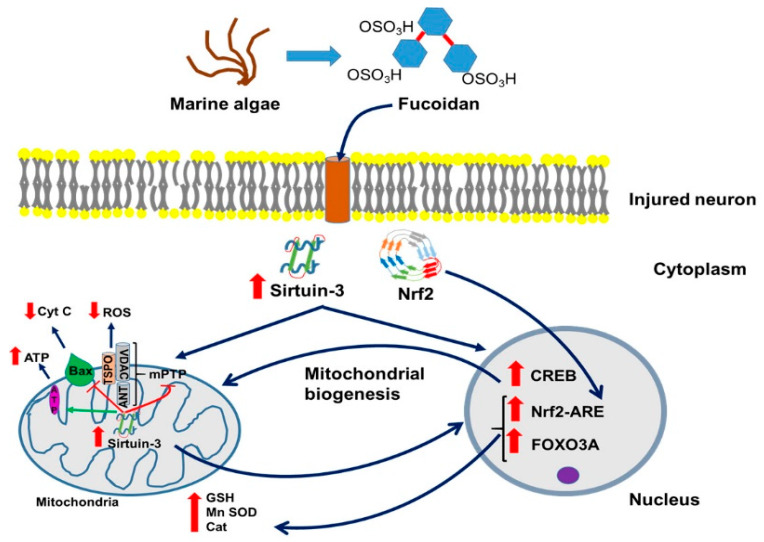
Mechanism of action of fucoidan in traumatic brain injury (TBI). Fucoidan alleviates brain injury through upregulation of sirtuin, which decreases reactive oxygen species (ROS) overproduction by inhibiting the mitochondrial permeability transition pore (mPTP) opening, and restores normal mitochondrial function via stimulation of ATP synthesis, and attenuates mitochondria-initiated apoptosis by decreasing leakage of cytochrome c from the mitochondria into the cytosol. Additionally, fucoidan stimulates expression of FOXO3A and Nrf-2-ARE genes, thus increasing glutathione (GSH) production and Mn-SOD and Cat activity.

**Figure 2 marinedrugs-18-00242-f002:**
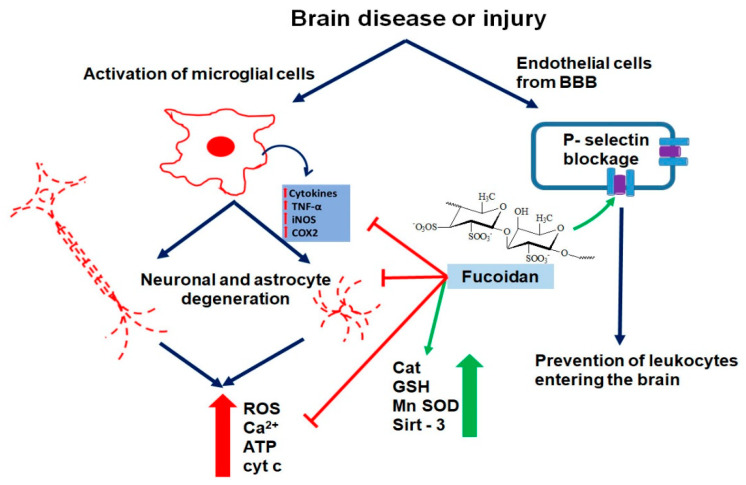
Effects of fucoidan on brain disease. Fucoidan reduces inflammatory response in brain diseases by inhibiting microglial activation, thus resulting in significantly decreased neuronal and astrocyte degeneration due to diminishing production of pro-apoptotic agents and improving antioxidant responses of the cell. Furthermore, fucoidan prevents leukocyte adhesion to the brain by blocking P-selectin.

**Figure 3 marinedrugs-18-00242-f003:**
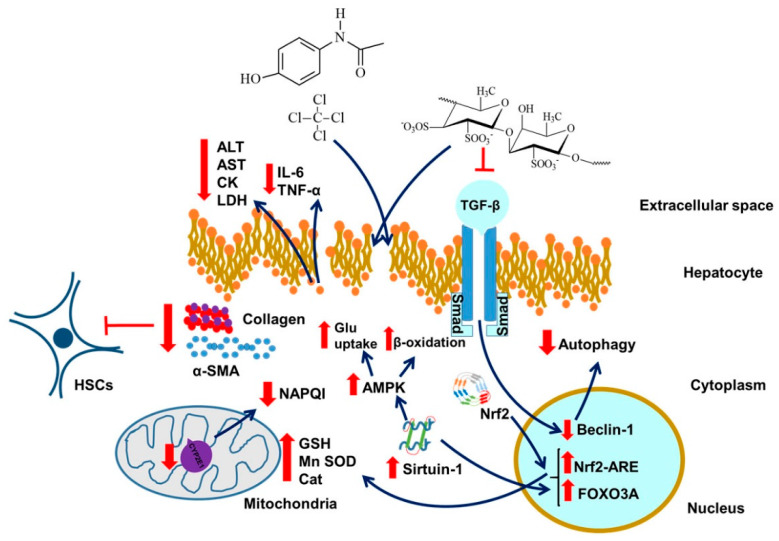
Effects of fucoidan on liver injury. Damaging agents at the top left side, APAP and carbon tetrachloride (CCl_4_), and protective fucoidan at the top right side. Fucoidan averts liver fibrosis by inhibiting HSCs production through optimal synthesis of collagen and alpha smooth muscle actin and prevents tissue damage by reducing transaminase release and restoring antioxidant potentials of cells. It decreases CYP2E1 activity, which reduces levels of toxic metabolites and inhibits TGF-β/Smad pathway, thereby hindering the occurrence of autophagosomes. Fucoidan also stimulates expression of sirtuin-1 in the liver, which activates AMPK and alleviates insulin resistance.

**Table 1 marinedrugs-18-00242-t001:** IC_50_ values of radical scavenging activity and chemical content of fucoidans isolated from dried algae and commercial supplements.

Specimen	DPPH^1^ Scavenging (mg/mL)	NO^2^ Scavenging (mg/mL)	O_2_^−^ Scavenging (mg/mL)	Fucoidan (%) ^2^	Sulfate Content (%)
Marinova Fucoidan	2.50 ± 0.18 ^a,3^	3.58 ± 0.33 ^a^	1.41 ± 0.38 ^a^	25.00 ± 0.03 ^a^	27.04 ± 0.92 ^a^
Daiso Fucoidan	4.10 ± 0.53 ^b^	5.76 ± 0.35 ^b^	3.83 ± 0.58 ^c^	4.26 ± 0.68 ^b^	2.89 ± 0.32 ^d^
*Porphyra* sp.	12.59 ± 1.13 ^c^	7.86 ± 0.12 ^c^	4.56 ± 0.51 ^c^	1.35 ± 0.17 ^c^	0.36 ± 0.06 ^e^
*P. tenera*	22.54 ± 2.68 ^d^	43.69 ± 6.18 ^d^	2.80 ± 0.33 ^b^	0.81 ± 0.01 ^d^	3.88 ± 0.26 ^c^
*U. pinnatifida*	42.77 ± 1.09 ^e^	34.17 ± 0.75 ^d^	2.29 ± 0.09 ^b^	2.14 ± 0.15 ^e^	5.48 ± 0.68 ^b^

^1^ IC_50_, Concentration of the sample at which the inhibition rate is equal to 50 %; ^2^ % Indicates g/100 g dry weight; ^3^ Results are presented as mean ± SD from three parallel measurements. Different small letters within a column represent statistical significance of *p* < 0.05 between the samples.

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
