# Peer review of "Potential Beneficial Actions of Fucoidan in Brain and Liver Injury, Disease, and Intoxication—Potential Implication of Sirtuins"

_marinedrugs, 2020, doi:10.3390/md18050242_

Round 1

Reviewer 1 Report

This review by Dimitrova-Shumkovska et al. illustrate that fucoidan is very effective in vitro and in vivo for brain injury, neuronal degeneration and liver injuries. However, have some comments for this manuscript listed as following:

  1. The fucoidans are sulfated polysaccharides present in brown marine algae. The author should explain how the fucoidans enter the blood circulation to brain and liver through the digestive system.
  2. The author must compare the differences between the fucoidans and LMWF. Are LMWF better than fucoidans on brain injury, neuronal degeneration and liver injuries?
  3. The authors should explain whether the molecule weight size and sulfated group of fucoidan affect their physiological activity.

Reviewer 2 Report

The manuscript entitled Potential beneficial action of fucoidan... is an interestingb review-article on the potential effects of fucoida a seawed derivative in several different pathological conditions affecting the brain and the liver. Besides the extesnive literature on the different model of neurodegeneration or stroke or even traumatic injuries, it ha sto be highlighted that these derivatives have a potential action on mitochondrial activity. It is an effect non related to a pathophysiological mechanisms, rather to a support to mitochondrial activity. In this view it would be useful in cases of metabolic encephalopaties rather than in degenerative disorders. Of note and interest is the anti-inflammatory activity useful in particular in stroke and most of all in brain injuries to contain the possible evolytion of tissue damage. I have no concerns except for more enphsis for metabolis brain disordres and less for neurodegeneration

Round 2

Reviewer 1 Report

The authors have revised the article according to the suggestions of reviewers.

So I have no comments and suggestions about this article.